REGISTERED REPORT PROTOCOL

# Comparison of clinical esthetic outcomes of immediately placed implants with and without immediate provisionalization in single-tooth implants of the esthetic area: A protocol for systematic review and meta-analysis

**Jianyong Dong**[1☉], **Chunmei Mao**[2☉], **Jie Xu**[3], **Yanting He**[3], **Kaiqi Zhang**[3], **Jun Cui**[3]*

**1** Gaoxin Branch, Jinan Stomatological Hospital, Jinan, Shandong Province, China, **2** Department of Oral Disease, Xinyi People's Hospital, Xinyi, Jiangsu Province, China, **3** Department of Dental Implantology, Central Laboratory, Jinan Stomatological Hospital, Jinan, Shandong Province, China

☉ These authors contributed equally to this work.
* cuijun516@126.com

## Abstract

### Background

Immediately placed implants with immediate provisionalization have become attractive options for patients and clinicians. However, there is no agreement on the esthetic advantages of immediately placed implants with immediate provisionalization. The aim of this systematic review and meta-analysis will be to assess the effect of immediate provisionalization on the clinical esthetic outcomes of immediately placed implants in a single-tooth implant of the esthetic area.

### Methods

An electronic search of MEDLINE/PubMed, EMBASE, Web of Science, Clinicaltrials.org., Cochrane Library, China Biology Medicine (CBM), Wan Fang Database, China National Knowledge Infrastructure Database (CNKI), and VIP Science Technology Periodical Database will be performed. The search will be carried out in the databases for articles published until May 2021. Two researchers will independently perform the literature selection, data extraction and quality assessment. Only randomized controlled trials (RCTs) reporting on the impact of immediate provisionalization on the clinical esthetic outcomes of implants immediately placed in a single-tooth implant of the aesthetic area will be included. The primary outcome of this study will be the esthetic outcome assessed by the objective index and patient satisfaction. The survival rate of implants and restorations and the changes in mucosa and bone around implants will also be analyzed. The included studies will be analyzed by Review Manager 5.3, and a meta-analysis will be performed.

**Data Availability Statement:** No datasets were generated or analyzed during the current study. Data will be made available after study completion.

**Funding:** This study was funded by the Dean's Research Assistance Foundation of Jinan Stomatological Hospital (No. 2018-03). The funders had and will not have a role in study design, data collection and analysis, decision to publish, or preparation of the manuscript.

**Competing interests:** The authors have declared that no competing interests exist.

## Results

The study will evaluate the clinical esthetic outcomes of immediately placed implants with and without immediate provisionalization in single-tooth implants of the esthetic area. The results will provide clinicians with a better treatment approach in their application.

## Conclusion

This systematic review and meta-analysis will provide more reliable, evidence-based data for the impact of immediate provisionalization on the clinical esthetic outcomes of immediately placed implants, which may or may not be beneficial.

## Registration number

PROSPERO registration number: CRD42021221669.

## Introduction

Immediate implant placement, in which the implant is placed directly after the extraction, can reduce the treatment time and simplify the surgical procedure [1]. Immediate implant placement cannot change the remodeling of alveolar bone after tooth extraction and may even result in the recession of mucosa [2–4]. The stability of peri-implant soft and hard tissue is the basis of satisfactory esthetic outcomes. The volume reduction of soft and hard tissue compromises a poor esthetic outcome. Many factors account for the recession of mucosa, including the position of the implant, the tissue phenotype, and the buccal plate thickness. Bone grafting, soft tissue augmentation, the flapless technique and other surgical strategies have been tested to preserve peri-implant tissue and improve esthetic outcomes in immediate implant placement [5]. Immediate implant placement with strict inclusion criteria can achieve predictable effects with low-risk esthetic outcomes [6].

Immediate esthetic rehabilitation is the key advantage of immediately placed implants with immediate provisionalization with high subjective satisfaction rates of patients. Immediate provisionalization can maintain the peri-implant soft tissues and restore the provisional esthetic outcome during the healing period. However, the effects of immediate provisionalization on the bone level, mucosal level and esthetic outcome for a long period remain controversial. Several studies have reported that compared with the delayed restoration of immediately placed implants, immediate provisionalization obtained more stable soft tissue levels, less bone volume reduction, and better esthetic outcomes [7–9], while other studies have shown that immediate provisionalization cannot improve esthetic outcomes [10–12]. In consideration of the limited sample sizes, the different surgical procedures and the varied observation periods of the studies, the conclusion is unclear. The inconsistency of these results prompted our study to examine the effect of immediate provisionalization on the clinical esthetic outcomes of immediately placed implants. We will conduct a systematic review and meta-analysis of randomized controlled trials (RCTs) published until May 2021. The aim of this study was to compare the difference in clinical esthetic outcomes of implants immediately placed with and without immediate provisionalization in single-tooth implants of the esthetic area.

## Materials and methods

The protocol has been registered in PROSPERO (CRD42021221669) at https://www.crd.york.ac.uk/PROSPERO/. The systematic review and meta-analysis will be performed according to

the Preferred Reporting Items for Systematic Reviews and Meta-Analyses Protocols (PRIS-MA-P) statement. Ethical approval will not be required, as this study will be based on aggregate data and will not involve humans.

The research question is as follows: Do immediately placed implants with immediate provisionalization improve clinical esthetic outcomes compared with immediately placed implants without immediate provisionalization of single-tooth implants in the esthetic area?

The PICOS framework will be used, and the elements will be as follows:

P (population): patients undergoing immediately placed single-tooth implants in the esthetic area [13].

I (intervention): immediately placed implants with immediate provisionalization.

C (comparison): implants without immediate provisionalization.

O (outcome): The primary outcome: the esthetic outcome and patient satisfaction. Secondary outcomes: the survival rate of implants and restorations; the changes in mucosa and bone around the implants including marginal bone level, marginal gingival level, plaque index, probing depth; the rate of peri-implantitis or peri-implant mucositis.

S (study design): randomized controlled trials.

## Data source and search strategy

Two reviewers (Jianyong Dong and Chunmei Mao) will search the following databases: MED-LINE/PubMed, EMBASE, Web of Science, Clinicaltrials.org., Cochrane Library, China Biology Medicine Database (CBM), Wan Fang, China National Knowledge Infrastructure Database (CNKI), and VIP Science Technology Periodical Database. The search strategy will include "immediately placed implants," "immediate provisionalization," "immediate restoration," "delayed restoration," "esthetic outcomes," and the Chinese key words corresponding to the words above. Both medical subject heading terms (MeSH) and free-text words will be used. The search will be performed for studies published until May 2021.

The literature search strategy will be created according to the Cochrane handbook guidelines. The PubMed search strategy were conducted on May 19, 2021 as follows:

#1 (immediate implant installation [Title/Abstract]) OR (immediate dental implant [Title/Abstract]) OR (Immediate implant placement [Title/Abstract]) OR (dental implantation [MeSH Terms]) OR (dental implantation, endosseous [MeSH Terms])

#2 (Immediate Dental Implant Loading [MeSH Terms]) OR (immediate loading [Title/Abstract]) OR (immediate restoration [Title/Abstract]) OR (immediate provisionalization [Title/Abstract]) OR (conventional restoration [Title/Abstract]) OR (conventional loading [Title/Abstract]) OR (conventional restoration [Title/Abstract]) OR (delayed loading [Title/Abstract]) OR (delayed restoration [Title/Abstract])

#3 #1 OR #2

#4 (Esthetics, Dental [MeSH Terms]) OR (esthetic outcomes [Title/Abstract])

#5 (randomized controlled trial [Publication Type]) OR (randomized controlled trials as topic [MeSH Terms]) OR (randomized controlled trials [All Fields]) OR (randomised controlled trials [All Fields])

#6 #3 AND #4 AND #5

An appropriate retrieval strategy will be adopted for the other databases.

## Inclusion and exclusion criteria

Studies will be included if they fulfill the following inclusion criteria: (a) RCTs about immediately placed implants with and without immediate provisionalization, (b) studies including sufficient information on peri-implant tissue change, (c) studies evaluating at least 10 patients with 12 months of follow-up, and (d) studies written in English or Chinese.

The exclusion criteria will be as follows: (a) cohort studies, case-control studies, case series, case reports, animal trials, in vitro studies, and systematic reviews; (b) studies involving the application of any additional therapy (soft tissue augmentation) that could have affected the healing outcomes; and (c) studies with insufficient data.

## Outcomes

Primary outcome indicators. The primary outcome of this study will be the esthetic outcome assessed by the objective index and patient satisfaction. The objective index of peri-implant esthetic outcomes will be determined using the Pink Esthetic Score/White Esthetic Score (PES/WES) [14]. The PES/WES comprises five variables of peri-implant soft tissue and five variables of the implant crown. A score of 2, 1, or 0 is assigned to all ten parameters, and a higher score means better esthetic outcomes. Patient satisfaction will be measured using a visual analog scale (VAS). The VAS questionnaire was completed by patients based on their overall satisfaction.

Secondary outcome indicators. Secondary outcomes will include the rate of implant/restoration survival and peri-implantitis/peri-implant mucositis and the changes in marginal bone level, marginal gingival level, plaque index, and probing depth.

## Study selection and data extraction

Two well-trained reviewers (Jianyong Dong and Chunmei Mao) will search the databases and select studies. They will screen the titles, abstracts and full text of the retrieved articles separately to identify articles that meet the review-eligibility criteria.

The data and other information regarding the included studies, including first author, publication year, study type, number of patients, sex and age of patients, physical condition of patients (diabetes, periodontal disease or other diseases affecting implants), number of smoking patients, number of implants, implant system, follow-up period, and primary and secondary outcome data, will be extracted. Disagreement between the two reviewers will be resolved by consulting a third professional reviewer (Jun Cui).

## Risk of bias assessment

The risk of bias assessment of the included RCTs will be independently performed using the Cochrane Collaboration's software program [15]. The two reviewers (Jianyong Dong and Chunmei Mao) will assess the risk of bias independently. The main domains will include random sequence generation, allocation concealment, blinding of participants and personnel, blinding of outcome assessment, incomplete outcome data, and selective reporting. The studies will be judged as having a low, unclear, or high risk of bias. Any disagreement will be resolved by consulting a third professional reviewer (Jun Cui).

## Assessment of heterogeneity

Statistical heterogeneity across studies will be assessed by the $I^2$ test and chi-square test [16]. If there is no statistical heterogeneity ($I^2 < 50\%$, $P > .1$), we will use a fixed effects model to conduct the analysis. Otherwise, a random effects model will be adopted.

### Data analysis and synthesis

Data analysis will be performed with Review Manager 5.3 software. If there is no heterogeneity observed among the studies, meta-analysis will be performed with Review Manager 5.3 software. Continuous data will be expressed as the mean difference (MD), and 95% confidence intervals (CIs) will be used. Dichotomous data will be expressed in risk ratio (RR) estimates, and 95% CIs will be used.

### Subgroup analysis

If significant heterogeneity is observed, subgroup analyses will be performed. Subgroup analysis will be used to explore potential sources of heterogeneity (such as age, sex, gingival biotype (thick or thin), bone grafting, use of the flapless technique and follow-up time).

### Sensitivity analysis

Sensitivity analysis will be conducted to examine the influence of individual studies on the quality of the meta-analysis results.

### Publication bias

Published bias will be evaluated by a funnel plot [17]. If there is an asymmetrical funnel plot, publication bias will be suspected. In contrast, publication bias will not be suspected if the points become symmetrically distributed on the funnel plot.

### Grading the quality of evidence

The quality of evidence will be assessed according to the Grading of Recommendations Assessment, Development, and Evaluation (GRADE) classification system [18]. On account of the risk of bias, imprecision, inconsistency, indirection, and publication bias, the articles will be divided into 4 levels according to the criteria as follows: very low, low, moderate, or high.

## Discussion

In addition to osseointegration and implant survival, the esthetic outcomes of immediately placed implants have received much attention. How to obtain a favorable esthetic outcome is challenging, particularly in the esthetic area [19]. Alveolar bone is vital for achieving satisfactory clinical esthetic outcomes. Immediately placed implants cannot prevent the physiological remodeling of the alveolar bone [20]. Immediately placed implants with immediate provisionalization may contribute to esthetic outcomes. Immediately placed implants with immediate provisionalization are technique sensitive [21]. The application of this technique not only requires advanced knowledge and experience but also careful case selection and strict inclusion criteria. Moreover, it is necessary to provide convincing evidence for the extensive use of immediately placed implants with immediate provisionalization through high-quality systematic review and meta-analysis of the existing RCTs about the issue. The protocol will provide a better understanding of the technique. The meta-analysis can objectively evaluate the technique and provide a reliable evidence-based basis for routine clinical application.

There are some limitations in the present study, including language limitations, a short follow-up period, and variable statistical data. RCTs with large sample sizes and long follow-up periods are needed to provide more powerful evidence for this issue. This systematic review and meta-analysis was performed according to the PRISMA statement and the Cochrane Collaboration Handbook.

## Supporting information

**S1 Checklist. PRISMA-P (Preferred Reporting Items for Systematic Review and Meta-Analysis Protocols) 2015 checklist: Recommended items to address in a systematic review protocol.**
(DOC)

## Author Contributions

**Data curation:** Jianyong Dong, Chunmei Mao, Jie Xu.

**Formal analysis:** Jianyong Dong, Chunmei Mao, Jun Cui.

**Funding acquisition:** Jianyong Dong.

**Investigation:** Chunmei Mao, Jun Cui.

**Methodology:** Jianyong Dong, Chunmei Mao, Jie Xu, Yanting He, Kaiqi Zhang.

**Software:** Yanting He, Kaiqi Zhang.

**Supervision:** Jun Cui.

**Validation:** Jianyong Dong.

**Visualization:** Jianyong Dong.

**Writing – original draft:** Jianyong Dong, Chunmei Mao.

**Writing – review & editing:** Chunmei Mao, Jun Cui.

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
