## [Decision Letter · Decision Letter 0]

28 Jun 2021

PONE-D-21-14369

Comparison of clinical esthetic outcomes of immediately placed implants with and without immediate provisionalization in single-tooth implants of the esthetic area: A protocol for systematic review and meta-analysis

PLOS ONE

Dear Dr. cui,

Thank you for submitting your manuscript to PLOS ONE. After careful consideration, we feel that it has merit but does not fully meet PLOS ONE’s publication criteria as it currently stands. Therefore, we invite you to submit a revised version of the manuscript that addresses the points raised during the review process.

Please disregard review comments saying that a research protocol is not needed and just take into account all comments that can be used to improve the quality of your manuscript.

We look forward to receiving your revised manuscript.

Kind regards,

Spyridon N. Papageorgiou, DDS, Dr Med Dent

Academic Editor

PLOS ONE

Journal Requirements:

"This study was funded by the Dean's Research Assistance Foundation of Jinan Stomatological Hospital (No. 2018-03)."

"The funders had and will not have a role in study design, data collection and analysis, decision to publish, or preparation of the manuscript."

4. Please upload a copy of Figure 1, to which you refer in your text on page 8. If the figure is no longer to be included as part of the submission please remove all reference to it within the text.

Reviewers' comments:

Reviewer's Responses to Questions

**Comments to the Author**

1. Does the manuscript provide a valid rationale for the proposed study, with clearly identified and justified research questions?

Reviewer #1: Partly

Reviewer #2: No

Reviewer #3: Yes

Reviewer #4: Yes

2. Is the protocol technically sound and planned in a manner that will lead to a meaningful outcome and allow testing the stated hypotheses?

Reviewer #1: Partly

Reviewer #2: No

Reviewer #3: Yes

Reviewer #4: Yes

3. Is the methodology feasible and described in sufficient detail to allow the work to be replicable?

Reviewer #1: No

Reviewer #2: No

Reviewer #3: Yes

Reviewer #4: Yes

4. Have the authors described where all data underlying the findings will be made available when the study is complete?

Reviewer #1: No

Reviewer #2: No

Reviewer #3: Yes

Reviewer #4: Yes

5. Is the manuscript presented in an intelligible fashion and written in standard English?

Reviewer #1: No

Reviewer #2: No

Reviewer #3: Yes

Reviewer #4: Yes

6. Review Comments to the Author

You may also provide optional suggestions and comments to authors that they might find helpful in planning their study.

Reviewer #1: The study plans to evaluate immediately placed single implants in the aesthetic region. However, the manuscript is not suitable and I am sorry to suggest a reject for its publication. Major problem with the study is that the results section is missing.

Language: The standard of English in the manuscript needs to be edited.There are many errors involving spelling and grammar. The manuscript would benefit from a thorough proof-read by a writer that has experience in writing/editing English based manuscripts for empirical peer reviewed science journals.

Abstract: I found abstract quite confusing, It should be a summary of the study. While in its present form, the authors mention how they had planned the work

• Introduction should be re-checked thoroughly for English language and fluency.

• First paragraph in my opinion, is not written well and should be written more clearly. There are many mistakes effecting the flow of the text. For example:

• RCT: Please explain this abbreviation in line 83 (which is used for the first time in text).

• Material- Methods should explain how the study was conducted, so it cannot be in future tense. Please apply corrections to this major problem with the text. It can be written: The study was planned as…, the work was conducted as… etc…

• I couldn’t understand the multiple parenthesis. Line 118,121

• Line 147: Primary outcome indicators. The primary outcome of this study will be the esthetic outcome assessed by the objective index and patient satisfaction.

What is objective index? Can authors explain what they mean, it is not clear if it is PES/WES or smt additional.

• Results section is missing

Discussions

• Discussions should be re-written according to the results.

Reviewer #2: This so-called protocol is not necessary, not at all. There are well established guidelines for systematic reviews for years and years, such as the well-known PRISMA guidelines. There is no need to establish search terms: these are logically connected to the subjected being reviewed. The proposal of local databases such as Chinese databases is inadequate, as this limits the authors that can search and read eligible articles from these, namely only those who speak the language. The same is valid for the inclusion criterion establishing Chinese as one of the languages. The esthetic outcomes are very well-established in the literature, as well as the ones for implant survival and for patient reported outcome measures. The guidelines for meta-analyses are also very well established. And not everyone has Review Manager as the preferred software to conduct meta-analysis. The same is valid for RR; some do prefer to calculate OD instead. Thus, this protocol is completely not necessary and absolutely does not add any novelty, not at all.

Reviewer #3: I had the opportunity to review the manuscript entitled “Comparison of clinical esthetic outcomes of immediately placed implants with and without immediate provisionalization in single-tooth implants of the esthetic area: A protocol for systematic review and meta-analysis.” and I recommend minor revisions to consider it for publication in the journal PLOS one.

Summary:

Suggestion:

The summary appear to be well written and readable. However, there are some key point to be addressed. First of all, Could the authors clarify how the risk of bias, such as age, gender and comorbidities will be handled?.

On the other hand, could authors explain what are the major contributions of this work?, since there different systematic reviews reported in literature covering the same topic?.

There are important characteristics of the patients that have not been considered in the document, representing risks of bias, such as age, gender, and comorbidities. I believe it is important to define how that information will be handled. On the other hand, in the literature, there are different systematic reviews reported on the subject. It is essential to describe what is the difference between those articles already published with this one?. What would be the new information that this document can provide?.

Issues:

The authors presented an organized and easy to follow manuscript. However, there are several issues which need to be addressed before this paper can be accepted for publication:

I recommend using MESH words for keywords, as an example: Immediate Dental Implant Loading.

Abstract:

1. I recommend specifying the objective at the end of the background in both the abstract and the full text.

2. In line 31 of the abstract and line 108 of the materials and methods, the MEDLINE and PubMed databases are indicated separately; considering PubMed is a repository of the MEDLINE database, I find it redundant.

3. I think it is necessary to highlight information from the meta-analysis in the abstract.

4. I suggest rewriting the abstract's conclusion because it has already been mentioned in the other sections above.

Background:

1. In line 83, the reader is informed that a systematic review will be carried out with RCTs from the last years. However, in line 35 of the abstract, the article reports that the information collected will be from inception until May 2021. I think it should be clarified.

Materials and Methods:

1. In line 103, it is reported that as part of the outcomes study is the plaque index; however, it is not mentioned in the outcomes item from line 146.

2. I consider it important to provide the date on which the search strategy was conducted.

3. Are there no considerations regarding criteria such as age, gender, or comorbidities in the inclusion and exclusion criteria? Healing can be altered with pathologies such as diabetes. How will the data of these patients be managed?

4. Considering that it is a study of evidence synthesis, do you think limiting the language would not alter the evidence?

5. The exclusion criteria should not be written in the article as the antagonistic sentence of the inclusion criteria (example: numerals a and d, between lines 140 and 144).

6. In the "Study selection and data extraction" section, the reviewers evaluate the titles and abstracts of the articles; however, the evaluation of the full text was not mentioned.

7. Regarding data extraction, I suggest specifying that the data from the primary outcomes, such as those related to esthetics, or secondary outcomes, such as satisfaction, survival, and plaque index, will be extracted. On the other hand, I have some questions: will the participants' characteristics not be extracted? Age and comorbidities will not influence the healing process? Will gender not affect the esthetic perspective?

8. In addition to the above, I recommend a subgroup analysis according to the outcome measures.

9. A publication bias assessment will not be performed?

10. I believe that it is appropriate to specify for the reader the type of evaluation that GRADE does.

11. I consider it essential to write the references of the tools used in the systematic review and meta-analysis methodology.

Discussion:

1. In the literature, there are different systematic reviews reported on the subject. What is the difference between this article and those already published?.

2. In line 215 of the discussion, the objective of the article was cited. I think it should be defined at the end of the background, and it is not necessary to explicitly mention it again in the discussion.

Reviewer #4: This protocol is well structured with a systematic review that addresses an interesting topic, a review registered in PROSPERO, PRISMA checklist, methodologically correct. But, there are some parts of the text wording that can be improved.

Abstract

Page 8, Line 35- "Each database will be searched from inception to May 2021." A clearer way of writing might be: The search will be carried out in the databases for articles published until May 2021.

Introduction

Page 10, line 83- “We will conduct a systematic review and meta-analysis of many RCTs published in the past few years…” As the search was not completed, I recommend removing the “many” and replacing “past few years” with “article published until May 2021”.

Materials and Methods

Page 8, line 31-“An electronic search of MEDLINE, EMBASE, PubMed, Web of Science,” – Authors should use the term: Pubmed/Medline, as the search will not be performed separately.

Page 11, line 98- "P (population): patients immediately placed single-tooth implants in the esthetic area." What are the references used by the authors in defining the aesthetic area? In the PROSPERO protocol, the authors mention using up to the second premolar, but premolars are posterior teeth, due to the force that affects these teeth.

Page 11, line 102- “O (outcome): implant survival rate, marginal bone level changes, marginal gingival level, plaque index, probing depth, the rate of peri-implantitis or peri-implant mucositis, esthetic outcomes, and patient satisfaction. ” In this section the author can determine which outcomes are primary and which are secondary.

Page 12, line 111- “The search keywords will include”, this term can be replaced by: The search strategy will include.

Page 13, line 142- "(b) studies involving the application of any additional therapy that could have affected the healing outcomes;" Determine what these therapies are.

Discussion

Page 17, line 219- “We hope the meta-analysis can objectively evaluate the technique”. I suggest the “we rope” exclusion.

Page 17, line 224- “We will conduct the systematic review and meta-analysis according to the PRISMA statement and the Cochrane Collaboration Handbook” I suggest writing it this way: This systematic review and meta-analysis is in accordance with the PRISMA statement and the Cochrane Collaboration Handbook

7. PLOS authors have the option to publish the peer review history of their article (what does this mean?). If published, this will include your full peer review and any attached files.

Reviewer #1: No

Reviewer #2: No

Reviewer #3: No

Reviewer #4: No

---

## [Author Response · Author response to Decision Letter 0]

24 Jul 2021

Dear Editors and Reviewers:

Thank you for your letter and the reviewers’ comments concerning our manuscript entitled “Comparison of clinical esthetic outcomes of immediately placed implants with and without immediate provisionalization in single-tooth implants of the esthetic area: A protocol for systematic review and meta-analysis”. (ID: PONE-D-21-14369).

These comments are all valuable and very helpful for revising and improving our paper, as well as the important guiding significance to our research. We have studied the comments carefully and have made corrections that we hope will be met with approval. Here, we did not list all of the changes, but the revised portion and main corrections are marked in red in the revised paper.

Thank you again for your time and consideration.

The point-by-point responses to the reviewers’ comments are as follows:

Reviewer #1:

Thank you very much for your comments and suggestions.

1. Response to comment: (The study plans to evaluate immediately placed single implants in the aesthetic region. However, the manuscript is not suitable and I am sorry to suggest a reject for its publication. Major problem with the study is that the results section is missing.)

Response: We apologize for not making this clear. The manuscript is just a protocol for meta-analysis, so we have not yet started to conduct the meta-analysis.

2. Response to comment: (Language: The standard of English in the manuscript needs to be edited. There are many errors involving spelling and grammar. The manuscript would benefit from a thorough proof-read by a writer that has experience in writing/editing English based manuscripts for empirical peer reviewed science journals.)

Response: In accordance with reviewer's comment, we have revised the whole manuscript carefully and tried to avoid any spelling or grammar errors. Additionally, the manuscript was edited for English language, grammar, punctuation, spelling, and overall style by several highly qualified native English-speaking editors at American Journal Experts. The certificate verification code was FDF2-0CB9-C0C8-F584-AADP at www.aje.com/certificate. All changes were marked in red text. Thank you again for this comment.

3. Response to comment: (Abstract: I found abstract quite confusing, It should be a summary of the study. While in its present form, the authors mention how they had planned the work)

Response: We apologize for the misunderstanding due to unclear descriptions in our manuscript. The manuscript is just a protocol for meta-analysis. We have not yet conducted the analyses, so we used the future tense.

4. Response to comment: (•Introduction should be re-checked thoroughly for English language and fluency.)

Response: Thank you for your advice. We have tried our best to revise the English of our manuscript. Additionally, the manuscript was edited for English language, grammar, punctuation, spelling, and overall style by the highly qualified native English-speaking editors at American Journal Experts.

5. Response to comment: (•First paragraph in my opinion, is not written well and should be written more clearly. There are many mistakes effecting the flow of the text. For example:

• RCT: Please explain this abbreviation in line 83 (which is used for the first time in text).)

Response: As suggested by the reviewer, we reedited our manuscript to make the it more clear and readable. We also defined “RCT” upon its first use in the text. Other changes are marked in our revised manuscript.

6. Response to comment: (•Material- Methods should explain how the study was conducted, so it cannot be in future tense. Please apply corrections to this major problem with the text. It can be written: The study was planned as…, the work was conducted as… etc…)

Response: The study has not been conducted, so we used the future tense in the manuscript.

7. Response to comment: (•I couldn’t understand the multiple parenthesis. Line 118,121)

Response: We have deleted the multiple parenthesis. Thank you for your suggestion.

8. Response to comment: (•Line 147: Primary outcome indicators. The primary outcome of this study will be the esthetic outcome assessed by the objective index and patient satisfaction.

What is objective index? Can authors explain what they mean, it is not clear if it is PES/WES or smt additional.)

Response: We apologize that the objective index was not addressed in detail. We have provided a further explanation of the PES/WES and added a reference to the manuscript.

9. Response to comment: (• Results section is missing

Discussions

• Discussions should be re-written according to the results.)

Response: We have not yet conducted the analyses, so the results section is missing. We have rewritten the discussion in accordance with your comments.

 

Reviewer #2:

Thank you very much for your comments and suggestions.

1. Response to comment: (There is no need to establish search terms: these are logically connected to the subjected being reviewed.)

Response: We only used the search strategy of PubMed as an example. It demonstrates that the protocol is practicable.

2. Response to comment: (The proposal of local databases such as Chinese databases is inadequate, as this limits the authors that can search and read eligible articles from these, namely only those who speak the language. The same is valid for the inclusion criterion establishing Chinese as one of the languages.)

Response: We searched Chinese databases and included related Chinese articles to decrease publication bias as much as possible.

3. Response to comment: (The esthetic outcomes are very well-established in the literature, as well as the ones for implant survival and for patient reported outcome measures. The guidelines for meta-analyses are also very well established. And not everyone has Review Manager as the preferred software to conduct meta-analysis. The same is valid for RR; some do prefer to calculate OD instead. Thus, this protocol is completely not necessary and absolutely does not add any novelty, not at all.)

Response: We conducted the protocol to increase the reproducibility of the results and address publication bias.

 

Reviewer #3:

We appreciate your constructive comments and suggestions. Your comments are valuable for improving our manuscript. We have made the changes in the manuscript in accordance with your comments. On behalf of my coauthors, we would like to express our great appreciation for your work and hope that the revised manuscript will be met with approval.

Once again, thank you very much for your good comments and suggestions.

Thank you and best regards.

Issues:

1. Response to comment: (Issues: I recommend using MESH words for keywords, as an example: Immediate Dental Implant Loading.)

Response: In accordance with your suggestion, we have rewritten the keywords in the revised manuscript.

Abstract:

1. Response to comment: (1. I recommend specifying the objective at the end of the background in both the abstract and the full text.)

Response: Following the reviewer’s suggestion, we have made corrections according to your comments.

2. Response to comment: (2. In line 31 of the abstract and line 108 of the materials and methods, the MEDLINE and PubMed databases are indicated separately; considering PubMed is a repository of the MEDLINE database, I find it redundant.)

Response: We apologize for our incorrect writing. We have made corrections in manuscript.

3. Response to comment: (3. I think it is necessary to highlight information from the meta-analysis in the abstract.)

Response: We agree the reviewer's good advice. We have made the modification in the abstract.

4. Response to comment: (4. I suggest rewriting the abstract's conclusion because it has already been mentioned in the other sections above.)

Response: Considering the reviewer’s suggestion, we have rewritten the conclusion in the abstract.

Background:

1. Response to comment: (1. In line 83, the reader is informed that a systematic review will be carried out with RCTs from the last years. However, in line 35 of the abstract, the article reports that the information collected will be from inception until May 2021. I think it should be clarified.)

Response: We are very sorry for our negligence. We have made this correction for consistency.

Materials and Methods:

1. Response to comment: (1. In line 103, it is reported that as part of the outcomes study is the plaque index; however, it is not mentioned in the outcomes item from line 146.)

Response: Thank you for the careful review. We have made the correction.

2. Response to comment: (2. I consider it important to provide the date on which the search strategy was conducted.)

Response: Thank you for your valuable suggestion. We have added the search date in the manuscript.

3. Response to comment: (3. Are there no considerations regarding criteria such as age, gender, or comorbidities in the inclusion and exclusion criteria? Healing can be altered with pathologies such as diabetes. How will the data of these patients be managed?)

Response: Thank you for your good comments. Age, sex, and comorbidities are important for aesthetic outcomes. We extracted the data of the included studies about age, sex, and comorbidities. If significant heterogeneity is observed, subgroup analyses will be performed. In the manuscript, we have taken those factors into consideration and made modifications.

4. Response to comment: (4. Considering that it is a study of evidence synthesis, do you think limiting the language would not alter the evidence?)

Response: Yes, your comments are right. However, for some foreign languages, we cannot understand the languages correctly. Our misunderstanding of languages may be the primary cause of incorrect results. Therefore, we only included Chinese and English articles. As you said, this may lead to certain publication bias.

5. Response to comment: (5. The exclusion criteria should not be written in the article as the antagonistic sentence of the inclusion criteria (example: numerals a and d, between lines 140 and 144).)

Response: We have fixed this as the reviewer suggested in our revised manuscript.

6. Response to comment: (6. In the "Study selection and data extraction" section, the reviewers evaluate the titles and abstracts of the articles; however, the evaluation of the full text was not mentioned.)

Response: Thank you for bringing the problem to our attention. We evaluated the full text of the articles. We have corrected the expression.

7. Response to comment: (7. Regarding data extraction, I suggest specifying that the data from the primary outcomes, such as those related to esthetics, or secondary outcomes, such as satisfaction, survival, and plaque index, will be extracted. On the other hand, I have some questions: will the participants' characteristics not be extracted? Age and comorbidities will not influence the healing process? Will gender not affect the esthetic perspective?)

Response: Your suggestions are important. According to your comments, we have rewritten this part.

8. Response to comment: (8. In addition to the above, I recommend a subgroup analysis according to the outcome measures.)

Response: Thank you for your suggestion. We will conduct subgroup analysis according to the outcome.

9. Response to comment: (9. A publication bias assessment will not be performed?)

Response: We have added information about publication bias to the revised manuscript.

10. Response to comment: (10. I believe that it is appropriate to specify for the reader the type of evaluation that GRADE does.)

Response: As the reviewer said, we have modified the part.

11. Response to comment: (11. I consider it essential to write the references of the tools used in the systematic review and meta-analysis methodology.)

Response: Many thanks for the reviewer’s suggestion. We have added the related references in the revised manuscript.

Discussion:

1. Response to comment: (1. In the literature, there are different systematic reviews reported on the subject. What is the difference between this article and those already published?)

Response: Thank you for pointing this out. We are not the first to conduct the review on the subject. Compared to the published reviews, we have two advantages: the enrolled articles were RCTs, and studies involving the application of any additional therapy (soft tissue augmentation) were excluded. We conducted this review to evaluate the effect of immediate provisionalization on the aesthetic outcomes of implants placed immediately. We have also rewritten the discussion according to your comments.

2. Response to comment: (2. In line 215 of the discussion, the objective of the article was cited. I think it should be defined at the end of the background, and it is not necessary to explicitly mention it again in the discussion.)

Response: We agree with the reviewer. We have made the modification.

 

Reviewer #4:

We appreciate your constructive comments and suggestions. Those details are important for improving the manuscript. We have made the changes in the manuscript according to your comments. We earnestly appreciate your work and hope that the corrections will be met with approval.

Once again, thank you very much for your comments and suggestions.

Thank you and best regards.

Abstract

1. Response to comment: (Page 8, Line 35- "Each database will be searched from inception to May 2021." A clearer way of writing might be: The search will be carried out in the databases for articles published until May 2021.)

Response: Thank you for your advice. We have made corrections according to the reviewer’s comments.

Introduction

2. Response to comment: (Page 10, line 83- “We will conduct a systematic review and meta-analysis of many RCTs published in the past few years…” As the search was not completed, I recommend removing the “many” and replacing “past few years” with “article published until May 2021”.)

Response: Following the reviewer’s suggestion, we have removed “many” articles and replaced “past few years” with “articles published until May 2021”.

Materials and Methods

3. Response to comment: (Page 8, line 31-“An electronic search of MEDLINE, EMBASE, PubMed, Web of Science,” – Authors should use the term: Pubmed/Medline, as the search will not be performed separately.)

Response: We apologize for our error. We have made this correction in the manuscript.

4. Response to comment: (Page 11, line 98- "P (population): patients immediately placed single-tooth implants in the esthetic area." What are the references used by the authors in defining the aesthetic area? In the PROSPERO protocol, the authors mention using up to the second premolar, but premolars are posterior teeth, due to the force that affects these teeth.)

Response: Thanks for the carefulness of the reviewer. As your suggestion, we would add the reference. The aesthetic area generally refers to the anterior maxilla, including maxillary incisors and maxillary canines. We apologize for our error regarding the PROSPERO protocol, and we have made the necessary correction.

5. Response to comment: (Page 11, line 102- “O (outcome): implant survival rate, marginal bone level changes, marginal gingival level, plaque index, probing depth, the rate of peri-implantitis or peri-implant mucositis, esthetic outcomes, and patient satisfaction.” In this section the author can determine which outcomes are primary and which are secondary.)

Response: Thank you for your advice. We have rewritten this part according to the reviewer’s suggestion.

6. Response to comment: (Page 12, line 111- “The search keywords will include”, this term can be replaced by: The search strategy will include.)

Response: Thank you for your advice. We have made the necessary corrections according to the reviewer’s comments.

7. Response to comment: (Page 13, line 142- "(b) studies involving the application of any additional therapy that could have affected the healing outcomes;" Determine what these therapies are.)

Response: Considering the reviewer’s suggestion, we have rewritten this part.

Discussion

8. Response to comment: (Page 17, line 219- “We hope the meta-analysis can objectively evaluate the technique”. I suggest the “we rope” exclusion.)

Response: Thank you for your advice. We have deleted “we hope” according to the reviewer’s comments.

9. Response to comment: (Page 17, line 224- “We will conduct the systematic review and meta-analysis according to the PRISMA statement and the Cochrane Collaboration Handbook” I suggest writing it this way: This systematic review and meta-analysis is in accordance with the PRISMA statement and the Cochrane Collaboration Handbook)

Response: As suggested, we have made the necessary corrections according to your comments.

We have tried our best to revise and improve the manuscript and made great changes in the manuscript according to the reviewers′ comments.

We look forward to your feedback about the revised manuscript, and we thank you for your good comments.

Thank you very much for your work on this paper.

We wish you all the best!

Yours sincerely,

Jun Cui

---

## [Decision Letter · Decision Letter 1]

6 Oct 2021

Comparison of clinical esthetic outcomes of immediately placed implants with and without immediate provisionalization in single-tooth implants of the esthetic area: A protocol for systematic review and meta-analysis

PONE-D-21-14369R1

Dear Dr. cui,

We’re pleased to inform you that your manuscript has been judged scientifically suitable for publication and will be formally accepted for publication once it meets all outstanding technical requirements.

Kind regards,

Spyridon N. Papageorgiou, DDS, Dr Med Dent

Academic Editor

PLOS ONE

Additional Editor Comments (optional):

Reviewers' comments:

Reviewer's Responses to Questions

**Comments to the Author**

1. Does the manuscript provide a valid rationale for the proposed study, with clearly identified and justified research questions?

Reviewer #2: No

Reviewer #3: Yes

Reviewer #4: Yes

2. Is the protocol technically sound and planned in a manner that will lead to a meaningful outcome and allow testing the stated hypotheses?

Reviewer #2: No

Reviewer #3: Yes

Reviewer #4: Yes

3. Is the methodology feasible and described in sufficient detail to allow the work to be replicable?

Reviewer #2: Yes

Reviewer #3: Yes

Reviewer #4: Yes

4. Have the authors described where all data underlying the findings will be made available when the study is complete?

Reviewer #2: No

Reviewer #3: No

Reviewer #4: Yes

5. Is the manuscript presented in an intelligible fashion and written in standard English?

Reviewer #2: Yes

Reviewer #3: Yes

Reviewer #4: Yes

6. Review Comments to the Author

You may also provide optional suggestions and comments to authors that they might find helpful in planning their study.

Reviewer #2: My opinion about the protocol has not changed. This protocol is completely not necessary and absolutely does not add any novelty. If PLOS One accept to publish this manuscript, it will be for exchange of money (the publication fee that is paid by the authors or their institution), because guidelines for systematic reviews are already well established for years. Instead of writing and submitting a protocol about "immediately placed implants with and without immediate provisionalization in single-tooth implants of the esthetic area", I could pretty well write very similar protocols about dental implant failure and the influence of smoking, or of bruxism, or of periodontal disease, or of radiotherapy etc. The main thing that would change is the terms of search. And I would "profit" by being the authors of multiple publications without doing much, and PLOS One would profit with the publication fee every time.

Reviewer #3: The authors applied the suggestions in the first review. Therefore, I consider that the document is suitable for publication.

Reviewer #4: The authors made the requested changes and the systematic review protocol is now clearer and easier to understand. Regarding the study methodology, the protocol is correct.

7. PLOS authors have the option to publish the peer review history of their article (what does this mean?). If published, this will include your full peer review and any attached files.

Reviewer #2: No

Reviewer #3: No

Reviewer #4: No

---

## [Editor Report · Acceptance letter]

8 Oct 2021

PONE-D-21-14369R1 

Comparison of clinical esthetic outcomes of immediately placed implants with and without immediate provisionalization in single-tooth implants of the esthetic area: A protocol for systematic review and meta-analysis 

Dear Dr. cui:

I'm pleased to inform you that your manuscript has been deemed suitable for publication in PLOS ONE. Congratulations! Your manuscript is now with our production department. 

Kind regards, 

on behalf of

Dr. Spyridon N. Papageorgiou 

Academic Editor

PLOS ONE